*J Physiol* 600.11 (2022) p 2815

## REPLY

### Reply from George A. Brooks

Langan and Navarro are thanked for providing commentary on our paper, 'Lactate in contemporary biology: a phoenix risen' (Brooks et al., 2022). We regard their thoughtful letter as a note in addition to our review on the lactate shuttle that is directly related to their provocative title: 'How high can the lactate phoenix rise?' (Langan & Navarro, 2022). Clearly, we do not know, but it is imperative to find the apogee.

Regarding the diverse signalling roles of lactate Langan and Navarro noted the role of lactate in activating the carotid body olfactory receptor (Olfr78) (Chang et al., 2015). Give the importance of breathing in physiology, ours was a significant omission making their contribution noteworthy. 'How high' indeed!

Langan and Navarro distinguished between the roles of lactate and hypoxia in hypoxia-inducible factor (HIF) signalling. We touched on that also in our review, but for us to have emphasized the role of lactate over hypoxia in HIF signalling might have been viewed as over reaching.

In our review we noted infancy of the field of histone lactylation on gene expression (Zhang et al., 2019). Already we see extensive evidence not only of histone, but protein lactylation (Leija et al., unpublished). Importantly, the role of lactate as a transcription factor was foreshadowed in an earlier report (Hashimoto et al., 2007).

In our review we briefly touched on the role of lactate as fuelling the spiral mitochondrial reticulum in sperm tails. Most recently, Sharpley et al. commented extensively on the role of lactate in embryogenesis (Sharpley et al., 2020). In terms of a launchpad for studies of lactate's role in biology, what could be more fundamental than the moment of conception and subsequent embryogenesis?

In sum, we thank Langan and Navarro for extending the discussion on the biological roles of lactate, particularly as a signalling molecule. As well, others have made similar observations, for instance the role of lactate in glutamatergic signalling in the brain (Pellerin & Magistretti, 1994). One significant correction we have to offer is to their penultimate sentence and citation on lactate as the primary circulating energy source. The paper cited was confirmatory; there having been many previous primary sources and reviews on lactate as an energy source (e.g. Brooks, 1985; Brooks, 2002).

George A. Brooks (ID)

*Department of Integrative Biology, University of California, Berkeley, CA, USA*

Email: gbrooks@berkeley.edu

Edited by: Ian Forsythe & Lykke Sylow

Linked articles: This is a reply to a Letter to the Editor by Langan & Navarro. To read the Letter to the Editor, visit https://doi.org/10.1113/JP283089. These Letters refer to the article by Brooks *et al*. To read this article, visit https://doi.org/10.1113/JP280955.

The peer review history is available in the Supporting Information section of this article (https://doi.org/10.1113/JP283189#support-information-section).

[Correction added on 1 June 2022, after first online publication: The copyright line was changed.]

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

## Additional information

### Competing interests

None.

### Author contributions

Sole author.

### Funding

NIH: R01 AG059715.

### Keywords

cardiac muscle, gluconeogenesis, lactate and brain, lactate shuttle, lactate signalling, lactylation, olfr78, oxidative metabolism

## Supporting information

Additional supporting information can be found online in the Supporting Information section at the end of the HTML view of the article. Supporting information files available:

**Peer Review History**

