## [Peer Review History · The Journal of Physiology]

Response to JP-LE-2022-283089R1 "How high can the lactate phoenix rise?" by Sean P. Langan and John S. Navarro

George A Brooks

DOI: 10.1113/JP283189

Corresponding author(s): George Brooks (gbrooks@berkeley.edu)

Review Timeline:

Submission Date:

22-Apr-2022

Accepted:

25-Apr-2022

Senior Editor: Ian Forsythe

Reviewing Editor: Lykke Sylow

Transaction Report:

Dear Dr Brooks,

Re: JP-RL-2022-283189 "Response to JP-LE-2022-283089R1 "How high can the lactate phoenix rise?" by Sean P. Langan and John S. Navarro" by George A Brooks

I am pleased to tell you that your Response has been accepted for publication in The Journal of Physiology.

NEW POLICY: In order to improve the transparency of its peer review process The Journal of Physiology publishes online as supporting information the peer review history of all articles accepted for publication. Readers will have access to decision letters, including all Editors' comments and referee reports, for each version of the manuscript and any author responses to peer review comments. Referees can decide whether or not they wish to be named on the peer review history document.

The last Word version of the paper submitted will be used by the Production Editors to prepare your proof. When this is ready you will receive an email containing a link to Wiley's Online Proofing System. The proof should be checked and corrected as quickly as possible.

All queries at proof stage should be sent to tjp@wiley.com

Yours sincerely,

Ian D. Forsythe
Deputy Editor-in-Chief
The Journal of Physiology
<https://jp.msubmit.net>
<http://jp.physoc.org>
The Physiological Society
Hodgkin Huxley House
30 Farringdon Lane
London, EC1R 3AW
UK
<http://www.physoc.org>
<http://journals.physoc.org>

REVIEWING EDITOR'S COMMENTS:

Thank you for providing a response to Dr Sean P. Langan and Dr John S. Navarro's commentary on your review, "Lactate in contemporary biology: a phoenix risen". We are happy to let you know that it has been accepted for publication.

Confidential Review

22-Apr-2022